# A Systematic Review on the Association of Acquired Human Cytomegalovirus Infection with Hearing Loss

**DOI:** 10.3390/jcm9124011

**Published:** 2020-12-11

**Authors:** Estrella Martinez-Gomez, Patricia Perez-Carpena, Marisa Flook, José A. Lopez-Escamez

**Affiliations:** 1Otology & Neurotology Group CTS 495, Department of Genomic Medicine, GENYO, Centre for Genomics and Oncological Research, Pfizer/University of Granada/Andalusian Regional Government, PTS Granada, Avenida de la Ilustración 114, 18016 Granada, Spain; percarpena@gmail.com (P.P.-C.); marisa.flook@genyo.es (M.F.); antonio.lopezescamez@genyo.es (J.A.L.-E.); 2Department of Otolaryngology, Instituto de Investigación Biosanitaria Ibs.GRANADA, Hospital Universitario Virgen de las Nieves, Universidad de Granada, 18014 Granada, Spain; 3Department of Surgery, Division of Otolaryngology, Universidad de Granada, 18016 Granada, Spain

**Keywords:** cytomegalovirus, sensorineural hearing loss, tinnitus, systematic review

## Abstract

Congenital cytomegalovirus (CMV) infection induces a clinical syndrome usually associated with hearing loss. However, the effect of acquired CVM infection in adults and children has not been clearly defined. The objective of this review is to critically appraise scientific evidence regarding the association of acquired CMV infection with postnatal hearing loss or tinnitus. A systematic review of records reporting sensorineural hearing loss (SNHL) or tinnitus and acquired CMV infection including articles published in English was performed. Search strategy was limited to human studies with acquired CMV infection. After screening and quality assessment, nine studies involving 1528 individuals fulfilled the inclusion criteria. A total of 14% of patients with SNHL showed evidence of previous exposure to CMV, while in individuals without SNHL (controls) the percentage rose up to 19.3%. SNHL was reported as unilateral or bilateral in 15.3%, and not specified in 84.7% of cases. The degree of SNHL ranged from mild to profound for both children and adults. None of the records reported tinnitus. The prevalence of children or adults with acquired SNHL with a confirmed acquired CMV infection by Polymerase Chain Reaction (PCR) or IgM anti-CMV antibodies is low. Phenotyping of patients with acquired CMV infection was limited to hearing loss by pure tone audiometry and no additional audiological testing was performed in most of the studies. Additional symptoms deserve more attention, including episodic vertigo or tinnitus, since some patients with the clinical spectrum of Meniere Disease could result from a CMV latent infection.

## 1. Introduction

The human cytomegalovirus (CMV) is a DNA virus included in the *herpesviridae* family, widely spread in the community. Subclinical infections do occur and they can involve all bodily organs, including the middle and the inner ear. Congenital CMV infection can produce a wide variety of clinical syndromes either by direct pathogenic effect or secondary to immune mechanisms. Acquired CMV infection is asymptomatic in most adults; however, it can produce severe symptoms in patients with compromised immune system [1].

Much of the pathogenesis associated with congenital CMV infection is explained by the virus’ ability to establish a latent infection in the host. In its latent state, the viral genome is maintained in the host cell without active replication; however, CMV is able to reactivate itself in response to changes in the host cell [2]. CMV persists for the lifetime of the infected host leading to a latent chronic infection, in which only a limited number of viral genes are expressed without genome replication. Cellular death pathways that are activated by viral infection as well as other biological processes, including control of the cell cycle and cellular stress responses, are efficiently regulated by CMV in the infected cell to facilitate virus replication. CMV has a broad cellular tropism and infects a large number of cell types during primary infection; however, the outcome of infection varies widely and is largely cell type-dependent and it may include endothelial, epithelial, fibroblasts, neuronal, monocytes/macrophages, granulocytes, and smooth muscle cells [3].

The histopathological damage of CMV disease was reported in 41 temporal bones of children with acquired CMV infection in a non-controlled study [4]. Typical inclusion bodies of CMV, namely multiple vacuoles with clusters of viral particles within the nucleus of affected cell was contained in the epithelium of the endolymphatic sac, the utricle, and the semicircular canals; there was loss of inner and outer hair cells, and of cochlear ganglion cells [5]. It has been postulated that many ear disorders, including Meniere’s disease (MD) [6], otosclerosis [7] and sensorineural hearing loss (SNHL) of various forms may have an underlying viral etiology. CMV could damage the inner ear leading to endolymphatic hydrops in immune-mediated inner-ear disease [8,9]. Moreover, it is plausible that CMV infection induces via pro-inflammatory state SNHL and tinnitus [10]. Among children with congenital CMV infection, the onset of deafness can occur in childhood even up to 6–8 years of life [8].

The seroprevalence of CMV differs greatly among different countries ranging from 50% up to 90% according to several factors, such as ethnicity or socio-economic status [11]. The prevalence of viral infection can be ascertained through various methods, including classical virological techniques such as transmission-electron microscopy, viral culture, or more recent methods offering comparably higher sensitivity, specificity, and throughput, sometimes broadly labeled as the ‘molecular-diagnostics’ of viral-infection [10]. Detection of CMV DNA can vary depending on the methods of DNA extraction [12], DNA amplification [13] and the part of the CMV genome being detected [14]. To confirm viral exposure in patients and control subjects, molecular ascertainments have been classified as those that aimed to (i) directly detect specific viral species through their nucleic acid or gene products (active or latent infection), or else (ii) indirectly detect specific viral exposures through antiviral antibody signatures of past or recent infection [10]. Prospective studies of CMV antibody including 2572 healthy blood donors have disclosed that between 40 and 100% of them have antibodies to CMV. Positive results of serologic tests presumably may reflect a latent congenital CMV infection [15,16].

While the precise mechanism of progressive hearing loss mediated by CMV infection is unknown, there are two possible mechanisms that would explain the clinical course: (1) the direct damage of hair cells or spiral ganglion neurons by a persistent acquired CMV infection, and (2) the host immune mediated CMV damage of infected cells [17]. The finding of CMV DNA in the inner ear fluids of children allowed to assume that this virus can not only injure the inner ear when acquired in pre-and perinatally age, but also later on [18] The main mechanism of SNHL in CMV congenital infection may be the alteration of potassium homeostasis via damage to the stria vascularis or Reissner’s membrane, with secondary damage to the organ of Corti. This mechanism can be also suggested for non-congenital infection since the virus is known to prefer undifferentiated cells (such as stem cells) [19].

The involvement of viruses in the pathogenesis of SNHL has been suggested by investigations conducted on animal models as well as from clinical, serologic, and histopathologic data collected on patients with acute idiopathic hearing loss. Several studies suggest that patients with SNHL, especially those with sudden onset [20], could be the result of a viral infection and *Herpesviridae* are currently considered the most likely etiological factor. A particular cochlear tropism of CMV has been reported in post-mortem inner ear of an immunocompromised patient who died from the complications of acquired disseminated infection by CMV [5].

There are many studies showing an association between congenital CMV infection and SNHL; however, the evidence to support an acquired CMV infection is limited. Most CMV infections have few symptoms and the inner ear damage associated with SNHL may be delayed, the causality being difficult to demonstrate.

The aim of the study is to critically appraise scientific evidence regarding the association of acquired CMV infection with postnatal hearing loss or tinnitus. For this, we conducted a systematic review and meta-analysis of case-control studies, which had employed molecular or serological ascertainments of CMV infection.

## 2. Experimental Section

This review was conducted using the Preferred Reporting Items for Systematic Reviews and Meta-Analyses (PRISMA) criteria [21] (Appendix A).

### 2.1. Search Strategy

On 20 October 2020, a literature search was conducted using three bibliographic databases (PubMed, Scopus and ResearchGate) to identify relevant peer-reviewed published studies. We did not consider abstracts without a full text. The selected keywords could appear in the title, abstract, text word, author keywords or MeSH Terms of the articles. If it was unclear whether an article should be included based on its title and/or abstract, the article was retrieved for full-text assessment. The keyword string used for literature search was: (“acquired cytomegalovirus” or “acquired CMV infection” or “adult CMV infection” or “children CMV infection”) AND (“sensorineural hearing loss” OR “tinnitus”). We did not apply date limits to our searches. In each database, we selected for original articles written in English, systematic reviews, case-reports and prospective or retrospective case-control studies. Only human studies and acquired CMV infection were included during literature search by configuring filters if available e.g., on PubMed.

The list of references was used to extract further publications. The filtered results were exported from each database and collated in reference manager software to remove duplicated records.

### 2.2. Research Questions and Selection Criteria

Several studies propose that SNHL or tinnitus could be associated with *Herpesviridae* infection. According to this hypothesis, we formulated the following research question: “Is acquired CMV infection associated with SNHL or tinnitus?” To answer this question, we followed a systematic approach (PICOS Questions):Population: patients (adults or children) with a clinical diagnosis of progressive or sudden SNHL or tinnitus.Intervention: individuals with SNHL or tinnitus were classified according to the evidence of acquired CMV infection. Individuals were defined as cases if they presented SNHL or tinnitus and they had direct or indirect evidence of acquired CMV infection. Direct methods included nucleic acid amplification Polymerase Chain Reaction (PCR) and optimization variants) or in situ hybridization, and indirect methods (IgM and IgG antiviral antibodies and Complement fixation antibodies). Individuals were considered as controls if they did not show any evidence of hearing loss, regardless of previous exposure to CMV.Comparison: absolute and relative frequencies of human acquired CMV infection in case and control groupsOutcome: audiological findings reported in individuals with acquired CMV infection.Study-design: case-controlled observational studies that ascertained a history of viral-infection with molecular-based diagnosis.

Selected records were classified according to method used to determine viral infection in case (progressive or sudden SNHL or tinnitus) or control groups. We distinguished between those methods that aimed to (i) detect the presence of specific viral species nucleic acids by PCR (during either active or latent infection) as opposed to (ii) detect evidence of infection at an indeterminate time through immunoglobulin antiviral antibodies. The former measurements are herein referred to as ‘direct’ (nucleid acid) and the latter as ‘indirect’ (serological). This distinction between ‘direct’ (nucleic acid) and ‘indirect’ (serological) ascertainment was important in the interpretation of the findings in the included studies.

Exclusion criteria:Human uncontrolled case series.Animal studies.Published studies in a language other than English.Congenital CMV infection.

### 2.3. Data Extraction and Variables of Interest

The data were retrieved from each study and the following information was extracted: first author’s last name, year of publication, study design, population studied, case and control group definition, hearing results, method of ascertainment and determinant of viral-infection and relationship between acquired CMV infection and SNHL (Table 1) [22,23,24,25,26,27,28,29,30].

### 2.4. Quality Assessment of Selected Studies

The selected records were screened to exclude review articles, meta-analysis, and irrelevant records (cases series and uncontrolled articles). The Cochrane Collaboration Tool was used to assess the scientific quality of each study [31]. (Appendix A). Next, the retrieved records were considered for full-text assessment. All selected records included a case and a control group. The variables of interests are shown in Table 1 [22,23,24,25,26,27,28,29,30].

### 2.5. Data Synthesis and Meta-Analysis

Data were extracted and the meta-analysis was conducted using GraphPad Prism v6.0 (GraphPad Software, San Diego, CA, USA). Unadjusted odds ratios (ORs) with 95% confidence interval (CI) were calculated to estimate the association between SNHL and CMV infection. Strength of association for the overall effects was determined through the Fisher’s exact test. Odds ratios underwent logarithmic transformation (‘logOR’) to display pooled and summary effects.

## 3. Results

### 3.1. Study Selection

Our search strategy yielded 411 non-duplicate articles. After screening the title and abstract of those articles, 371 records were excluded for irrelevance. The discarded records were non-human studies or abstracts presented at scientific meetings. After the full-text screening of the 40 remaining articles, a further 31 were excluded for the reasons detailed in Figure 1. Thus, after assessment, 9 full-text studies satisfied our eligibility criteria and were included for sample description, critical appraisal, and data synthesis. These studies were published over a period of 30 years, between 1985 and 2015.

### 3.2. Sample Description and Audiological Findings

A total of 1528 individuals were included in the nine selected records according to the inclusion criteria. Among the 1528 individuals included, 628 subjects suffered from some type of hearing disorder: uni- or bilateral hearing loss (mild, moderate, severe or profound) or sudden SNHL.

All patients were evaluated by pure-tone audiometry to confirm hearing loss. Furthermore, auditory brainstem evoked potentials to assess auditory neuropathy was reported in one study. The studies showed several differences among them, including the definition of case group, the ethnic background of the population and several differences in the methods.

Most of the studies were conducted in children of few years of age, although it was observed that patients with acquired CMV infection (children or adults) could potentially present hearing loss independently of the age group. The records were classified according to the age of the included patients, as children (age below 14 years old; *n* = 1172) and adults (age above 14 years; *n* = 139). In one study, this information was not specified (*n* = 217) [30]. Among the 1528 individual (628 SNHL patients and 900 controls), only 92 suffered from SNHL after acquired CMV infection; however, none of 174 individuals with acquired CMV infection presented SNHL (Figure 2).

A total of 14% of patients with SNHL were positive for acquired CMV infection, while in individuals without SNHL (controls) the percentage increased up to 19.3%. The rate of CMV infection was higher in controls than in patients. All articles reported the type of hearing disorder, being as unilateral or bilateral in 15.3%, and not specified in 84.7% of SHNL cases. Only two studies [26,28] reported the degree of hearing loss that ranged from mild (0.8%) or moderate (2.9%) to profound (12.3%) for both children and adults. None of the patients reported tinnitus.

### 3.3. Methods for the Diagnosis of Human Acquired CMV Infection

Several methods were used for the diagnosis of acquired CMV infection in patients with SNHL in the nine selected studies. Three studies directly ascertained viral-infection through nucleic acids detection, and all of them through PCR-based methods and five through virus culture. Four studies also ascertained viral infection through host antiviral immunoglobulin (three of these studies measured these antibodies through enzyme-linked immunosorbent assay (ELISA) and only one through a Complement Fixation assay (FCA).

### 3.4. Critical Appraisal

Across all nine studies, a total of 628 patients were compared to 900 controls. While all studies reported that SNHL was excluded in the control group, only three studies reported enough information to consider healthy participants as controls; another three studies considered CMV infection-negative patients as controls. These studies stated that their participants were healthy beyond merely being free of a history of inner-ear disease. All individuals selected as controls have not experienced auditory or vestibular symptoms, but it was not made explicit whether controls were known to have any chronic diseases.

### 3.5. Data Synthesis and Meta-Analysis

A determinant of CMV-infection was specified within the methodology in 9 studies, and 6 of them presented extractable data for pooling. No significant association was found between CMV and SNHL (OR = 1.3 (0.93 to 1.8), Figure 3).

## 4. Discussion

The first report describing an association of acquired CMV infection with SNHL was published 40 years ago; however, the pathogenic mechanism of hearing loss is not fully understood. Our systematic review shows that acquired CMV infection demonstrated by PCR or ELISA is not significantly associated with SNHL in adults or children and the evidence to support an association between acquired CMV infection and SNHL is weak.

We have identified nine case-control studies to assess the association between acquired CMV infection in adults or children and SNHL. However, only six studies resulted appropriate to calculate the pooled OR. Among these studies, the frequency of SNHL in individuals with acquired CMV infection was slightly higher (19.78%) compared to the frequency of CMV infected individuals without SNHL (15.86%). However, given that only six studies could be selected for the metanalysis and the study of Verbeeck et al. [25] contributed with 48% of the individuals, we cannot rule out a bias. Overall, according to the meta-analysis, CMV shows no association with SNHL, while the association with tinnitus has not been investigated. For viral infection diagnosis, PCR and ELISA tests were the most used methods in these reports. The variability of the audiological profile of the CMV infected patients can result from a broad possibility of causes of hearing loss. Among them: the clinical condition of the patients, making them susceptible to numerous opportunistic infections, ototoxic drugs, and the action of the virus itself.

Four of the selected studies reported hearing thresholds. Two studies [22,30] described the same hearing thresholds (<40 dB) in case and control groups. However, the other two studies [24,28] showed differences among the case (mean 78 dB) and control groups (mean 60 dB). Paryani et al. reported a high frequency SNHL (>4000 dB) in all cases. Unfortunately, none of the studies included information about the age of onset of SNHL.

An examination of human temporal bones along with studies of animal models indicates that CMV (or a CMV protein) may be present in the sensorineural epithelium and the spiral ganglion neurons. This may cause damage to the inner ear by virus-mediated damage to hair cells, supporting cells or neurons, and/or secondary to host-derived inflammatory responses to CMV in the ear, resulting in injury to the cochlea and hearing loss [15]. However, this hypothesis does not fully explain the progressive and late-onset nature of CMV-related hearing loss. How CMV is related to the pathogenesis of SNHL is not known, and limited audiological assessment in patients with confirmed congenital CMV infection has been performed in most studies to determine if the primary damage occurs at the organ of Corti or the auditory nerve fibers. Moreover, CMV may affect the inner ear by several mechanisms that might include: (a) labyrinthitis secondary to a viremia, (b) labyrinthitis or neuritis secondary to meningitis, (c) a cranial polyneuropathy, (d) reactivation of a latent infection in spiral ganglion neurons, or e) alteration of the cellular immune response [3].

As a result of molecular mimicry between viruses and inner ear tissues, humoral and/or cellular immune responses in the endolymphatic sac or the cochlea could cause auditory and/or vestibular pathologies. Mechanisms for inner ear injury may be influenced by temporary alterations in cellular immunity secondary to simultaneous viral infections as well as the virulence of CMV. To define a virus as an etiologic agent for hearing loss, three criteria should be satisfied: (1) successful isolation of the virus from the perilymph or endolymph, (2) morphological identification of the virus in cochlear cells by electron microscopy or by characteristic cytopathological changes, and (3) detection of viral antigen in the cochlear cells by immunohistochemical analysis [12]. These criteria are difficult to fulfill in human cases, as hearing loss is not fatal and the encasement of the inner ear in dense temporal bone makes it difficult to study in vivo. Collecting appropriate samples from the inner ear of a living subject for virus isolation purposes would require considerable technical skill and has a low (but significant) risk of permanent deafness. Recently, cochlear implants have become the established therapeutic method for treating SNHL, and it is now possible to collect perilymph at surgery [17].

Acquired CMV infection of the inner ear might be associated with SNHL, but evidence to support causality in the pathogenesis of hearing loss in children or adults is limited. The cochlear lesions in congenital CMV infection are diffuse, but predominate in the stria vascularis in a mouse model [32]. The stria vascularis is one of the most highly vascularized covering and lining epithelia found in mammals and the only one containing intra-epithelial vessels. The CMV may therefore penetrate the inner ear by this route and infect the marginal epithelial cells [19]. Nevertheless, histopathological studies are limited in adults [33].

Active infection and inflammation were also found in the saccule and utricle, predominantly in the non-sensory epithelial cells such as the transitory cells and dark cells. Active CMV replication was noted in the inner ear epithelial cells, mesenchymal tissue, and bone marrow cells, but not in the highly differentiated neurons or sensorineural cells, according to experimental data [33]. In any case, establishing whether a virus (or another infectious agent) is relevant in the pathophysiology of a neurotological condition involves a claim beyond the plausibility of virulence and a significant association [34].

Some data show that congenital CMV-related hearing loss is associated with ongoing viral replication in the inner ear several years later after birth [35]. However, some reports have investigated the relationship between congenital CMV infection and hearing impairment. The discovery of viral DNA in the perilymph of three of the subjects enrolled in these studies provided an in vivo demonstration that Herpesviruses could be isolated in the inner ear when it occurs in postnatal age [36,37].

There is an ongoing debate about the etiology and pathophysiology of SNHL in adults. In particular, an extensive disagreement exists regarding the role of different viruses in the onset of idiopathic sudden SNHL. The most accepted cause of SNHL, regardless of the patient’s age, is viral labyrinthitis [30]. The finding of a prior viral infection in as many as 30–40% of SNHL patients presumably support this statement. However, several studies using conventional viral diagnostic tests have not confirmed that a viral infection is a common cause of SNHL [38]. According to the limited evidence, 60% of individuals without SNHL have antibodies anti-CMV IgG [39].

## 5. Limitations

Although the search strategy was designed to include all available studies on CMV infection, we selected case-controlled studies published in English. Our systematic review, although it is limited to nine studies, shows no evidence of viral infection by PCR or serological tests. Since the study of Verbeeck et al. [25] represents almost 49% of all individuals, the result of this meta-analysis is biased by the weight of this study. These results do not support acquired CMV infection as a main factor to develop SNHL in children or adults. However, the lack of appropriate controls in some studies raise particular concerns about their findings, and the fact that very few studies used direct and indirect ascertainment methods simultaneously to confirm acquired CMV infection questions the available evidence to support CMV as an etiological factor for SNHL in both children and adults. Evidence of acquired CMV infection (direct and indirect) is not time-stamped, so, to determine causality with SNHL is necessary to examine the prevalence of past exposure in individuals with hearing loss vs. that of the general population.

Although the association of hearing loss and acquired CMV infection was the main focus of this review, other associated manifestations were also observed in adults. Tinnitus was reported in one study [40], delayed endolymphatic hydrops [17] and immune-mediated inner ear disease in 56 individuals [41]. Moreover, isolated profound SNHL reported as “single sided-deafness” was reported in 24% of children and 21% of adults [17]. Nonetheless, it is not possible to establish an association between acquired CMV infection and vestibular disorders. The lack of a control group in these records limits the evidence to support any etiological role of the virus in the genesis of these manifestations.

On the other hand, phenotyping of patients with acquired CMV infection is limited to hearing loss assessment by pure tone audiometry, and additional symptoms deserve more attention including episodic vertigo or tinnitus since some patients with clinical spectrum of MD could result from a CMV latent infection.

## 6. Conclusions

There is no evidence to support an association between acquired CMV infection and SNHL, neither in adults nor in children. A large prospective cohort study in adults and children with confirmed CMV infection is needed to demonstrate the potential role of acquired CMV infection in the development of hearing impairment.

## Figures and Tables

**Figure 1 jcm-09-04011-f001:**
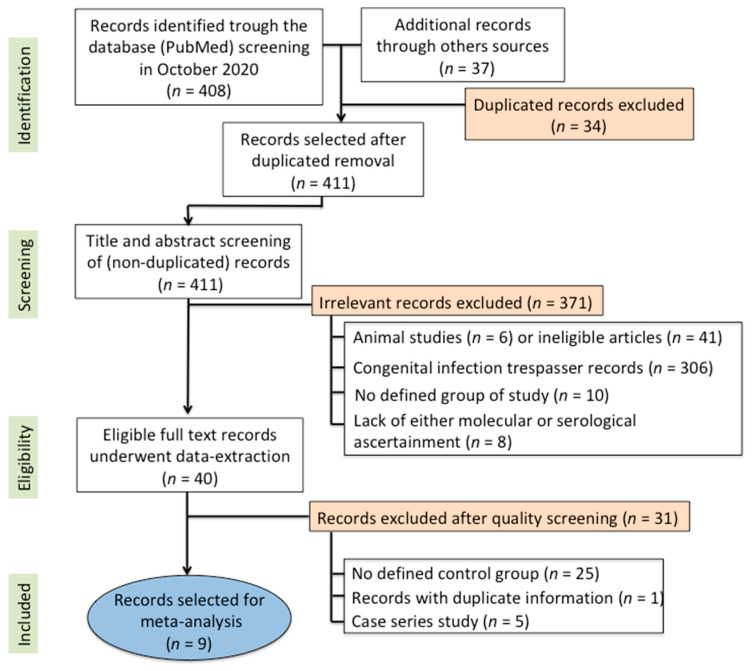
Flowchart of the study design and record selection for synthesis.

**Figure 2 jcm-09-04011-f002:**
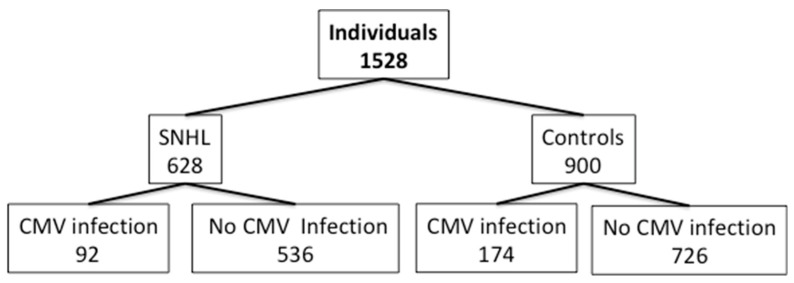
Total number of patients with sensorineural hearing loss and control classified according to the presence of acquired cytomegalovirus (CMV) infection (tested by Polymerase Chain Reaction (PCR)) and enzyme-linked immunosorbent assay (ELISA) or CF antiviral IgM).

**Figure 3 jcm-09-04011-f003:**
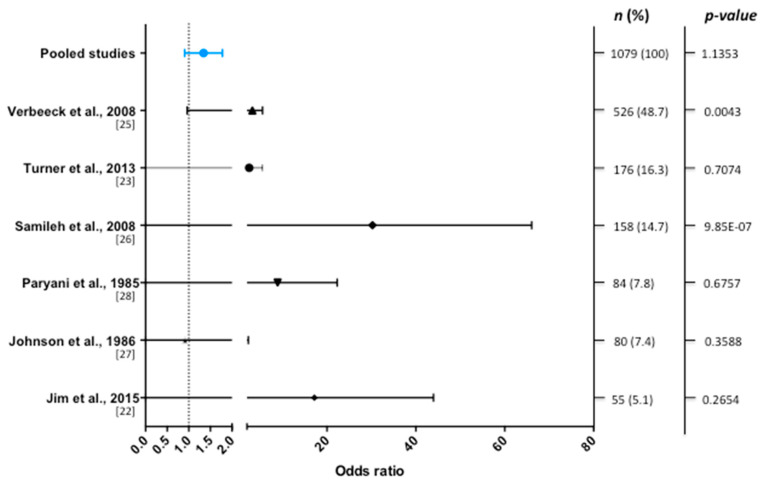
Forest plot of the selected studies showing odds ratios with 95% confidence intervals. Pooled data (blue line) from 1079 individuals show no significant association between CMV infection and hearing loss.

**Table 1 jcm-09-04011-t001:** Summary appraisal of the case-control included studies.

	Study Design	Population Studied (*n*)	Case Group Definition	Control Group Definition	Hearing results		Diagnosis of cytomegalovirus (CMV) infection		Relationship CMV- Sensorineural Hearing Loss (SNHL)
					Pure Tone Audiometry	Neurophysiological Testing	Ascertainment Method	Determinant	
Jim et al., 2015 [22]	Prospective case-control	Children <35 weeks (55)	Postnatal CMV infection through breast feeding (14)	Negative CMV infection (41)	Two infants with mild peripheral hearing impairment in both groups	No permanent delayed speech	Polymerase Chain Reaction (PCR) CMV isolation enzyme-linked immunosorbent assay (ELISA)	PCR-positive viral DNA in urine Culture positive urine (viruria) Antiviral IgM (serum)	None of the infants had CMV-related death or permanent SNHL
Turner et al., 2013 [23]	Retrospective case-control (from 1993 to 2008)	Infants with birth weights <1500 g (176)	Acquired CMV infection (16)	Negative CMV infection (160)	Suspected SNHL in 20% of infants with acquired CMV and 14% controls	Acquired CMV was not associated with neurologic sequelae	CMV isolation	Culture positive urine	Congenital CMV in infants is associated with high rates of neurologic injury and hearing loss comparing with acquired CMV infants
Kikidis et al., 2010 [24]	Prospective case-control	Adults patients with sudden SNHL (84)	CMV, herpes simplex virus, toxoplasma and Epstein-Barr infection (8)	Negative virus/toxoplasma infection (76)	Hearing level was 86.5 dB HL in the case group and 60.7 dB HL in the control group.		ELISA	Antiviral IgM (serum)	Recent subclinical viral or toxoplasmosis infections may be involved in the pathogenesis of sudden SNHL (in approx. 10% of cases), suggesting that sudden SNHL is not a single disease
Verbeeck et al., 2008 [25]	Retrospective case-control (from 2002 to 2004)	Babies born between April 2002 and April 2004 (526)	194 babies with indicative hearing impairment (22 positive CMV; 172 negative CMV)	332 matched controls without hearing impairment (15 positive CMV; 317 negative CMV)	Hearing impairment at birth was confirmed by an audiological center in 136 out of the 526 tested babies while 390 tested babies proved to have normal hearing.	None of the children had developed delayed-onset hearing loss at the end of the study	PCR CMV isolation	PCR-positive viral DNA in urine Culture positive urine	Significantly more babies with confirmed hearing impairment were CMV positive after birth Babies with CMV viral loads above 4.5 log copies/mL urine seem to be 1.4 times more likely to have confirmed hearing impairment.
Samileh et al., 2008 [26]	Case-control study (from 2002–2003)	Children <14 years (158)	Children with SNHL (95) Acute infection (CMV-IgM) in 33/95, previous immunity (IgG) in 69/95	Children without SNHL (63) Acute infection (CMV-IgM) in 2/57, previous immunity (CMV-IgG) in 54/57	Severity of SNHL in cases included: 80% profound >95 db; 15% moderate; 5% mild. Unilateral SNHL 22% (20/95); bilateral SNHL 78% (75/95).		ELISA	Antiviral IgG and IgM (serum)	CMV is one of the most common infectious agents in SNHL children compared to the healthy children. Probably both congenital and acquired CMV can induce progressive hearing loss.
Johnson et al., 1986 [27]	Prospective case-control	Infants (80)	Premature or sick term infants with perinatally acquired CMV (40)	Matched control subjects (40)	One case group subject had a bilateral SNHL> 4000 Hz. Four control subjects had SNHL, 3 requiring binaural hearing aids.	There were 12 transient conductive hearing losses due to serous or suppurative otitis media. Five of these losses were in the case group and seven were in the control group	CMV isolation	Culture positive Urine and/or saliva	Perinatally acquired CMV infection is not associated with significant SNHL in premature or full term infants through age 3.
Paryani et al., 1985 [28]	Prospective case-control	Infants (84)	Premature or sick term infants with perinatally acquired CMV (42)	Matched control subjects (42)	SNHL in 4 of 42 CMV infected patients (all mild-moderate) and in 2 of 42 controls (one severe)	All of the CMV infected patients had SNHL >4000 Hz, and hearing aids were not required.	CMV isolation	Culture positive urine and saliva	Acquired CMV infections are not associated with a significant increase in SNHL
Paradowska et al., 2013 [29]	Prospective case-control	Children (93) Adults (55)	Postnatal CMV infection + SNHL	CMV infected adults	Hearing loss in 10 children (11%)		Nested PCR (nPCR)	Viral DNA from peripheral blood, urine, and/or cerebrospinal fluid samples	No association was found
Wilson, 1986 [30]	Prospective case-control	Adults and children (217)	SNHL	Healthy patients	Profound and mild, mid-frequency hearing losses		Complement fixation assay (CFA)	Antiviral IgG and IgM (serum)	No differences were found between the effects of herpes virus infection upon the degree of hearing loss in patients with or without herpes infection

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
