# Peer review of "A Systematic Review on the Association of Acquired Human Cytomegalovirus Infection with Hearing Loss"

_jcm, 2020, doi:10.3390/jcm9124011_

Round 1

Reviewer 1 Report

The objective of the manuscript is to evaluate the possible existence of sensory neural hearing loss in post natal CMV infection through a literature review.

The methodology although correct is somewhat rendered biased by the information included in each or the different studies: viral testing in older studies (viral culture)  are not as sensitive as more recent studies.

Although the mechanism of an eventual hearing loss consecutive to post natal CMV infection is not known, the physiopathology of hearing loss to congenital CMV infection has been described: please refer to : Teissier N. Inner ear lesions in congenital cytomegalovirus infection of human fetuses. Acta Neuropathol. 2011 Dec;122(6):763-74. 

The viral tropism particularly targets cells of the stria vascularis and the supporting cells, both particularly important in potassium recycling in the inner ear. In case of postnatal infection, the viral tropism should be the same since the virus is known to prefer undifferentiated cells (such as stem cells instead of neurons in the brain). The viral presence in the spiral ganglion is noted although rarely found.

Therefore, these elements should be corrected in the text.

More emphasis should be put on the conclusion than no data supports that postnatal CMV infection leads to SNHL; some of the phrases are too nuanced and lead to think that it might still be possible, although data suggests the opposite.

Référence 31 l 264, p 4: probably incorrect; 

Reviewer 2 Report

Introduction

While the Abstract is well conceived, the objective of the review is not adequately explained in the introduction, as well as the difference between congenital and acquired CMV infection in not discussed. It sounds like a store of confuse ideas about the detection of various CMV signs. The introduction should be re-organized following a precise scheme: try to give more details about congenital and acquired CMV infection (clinical and serological diagnosis), then focus the question on audiology symptoms of CMV infection (especially infant hearing loss whether progressiveor not),  and finally point out the aim of the study.

Results

Lines 173-176. The meaning is completely unclear.

Discussion

Discussion is better organized than Introduction, but the main conclusion of the review (“There is no evidence to support a relationship between acquired CMV infection and SNHL, neither in adults nor in children”) has to be read between the lines. You are invited to add some other evidences and enlarge you discussion section.

line 223: ” The variability of the audiological profile of the CMV infected patients can result from a broad possibility of causes of hearing loss. Among them: the clinical condition of the patients, making them susceptible to numerous opportunistic infections, ototoxic drugs, and the action of the virus itself. Therefore, the selected studies did not  report an audiological profile in acquired CMV infected patients”.

Audiological data reported in the studies included deserve more relevance: try to give more details about auditory thresholds, age at HL onset, etiopathogenesis. Your discussion could obtain strength in this way.

268-271: there is too confusion about congenital and acquired infection and reference do not reflect what is discussed. Try to clarify the results of the mentioned studies.

279-280: probably some words are missing in the sentence because the meaning in the context is unclear.

Limitations

It is worth adding a paragraph named LIMITATIONS  that should contain from line 279.

It should be also mentioned that the results are based on a relatively small number of studies.

References

Reference are obsolete by now (2020). Please provide an update. Literature is full of recent works that you forget to mention, especially in Introduction and Discussion.

Language

There are problems with the English and grammar throughout.  I would expect a much higher standard of proof-reading and ideally checking by a native English speaker (this might be useful) for any final submission.

Minor

Page 1, line 22. Please rephrase and make clear the meaning of the sentence  “Search strategy was limited to human 22 studies, acquired CMV infection”

line 26. Please pay attention to English grammar and rephrase “SNHL was reported as unilateral or bilateral in 15,3%, and no specified in 84,7% of cases 26 and the degree of hearing loss that ranged from mild to profound for both children and adults”

line 29. What do you mean with “phenotyping?” Is it a noun or a verb?

Page 3

Lines 122- 128: please correct English grammar.

Page 8

Line 219-22: please correct English grammar.

Results: the verb suffer needs to be followed by “from” if you are talking about diseases.

Reference

I think that reference 31 belongs to line 262 and should not be mentioned two times.

Reviewer 3 Report

Abstract

CMV and PCR in the abstract – please introduce first

Introduction

The introduction does not convinced me that we need this study. There is lack of justification for the study.

The authors say: “The objective of this systematic review is to assess the evidence that acquired CMV infection could be involved in hearing loss or tinnitus.” This is not convincing. Anyone can just say that he wants to review anything. Why do we need such review? Why connection between CMV infection and hearing loss could be expected? Why it is important? Why mention tinnitus? There are many disorders related to hearing loss. Why the authors mentioned specifically the tinnitus?

Experimental section

There are some chaotic sentences in the Experimental section that provide more justification for the study than was presented in the Introduction. I suggest to move them there and focus here only on the methodology.

Discussion

First the authors state that “evidence to support a relationship between acquired CMV infection and SNHL is weak” then they devote the discussion to explain what mechanisms may be responsible for connection of CMV infection and SNHL.

Line 270: “However, some reports have investigated the relationship between congenital CMV infection and hearing impairment [33, 34].” – this sentence does not fit, it looks like introduction to something which does not follow. There is no continuation on the results on relationship between congenital CMV infection and hearing impairment.

General remarks:

It is good that authors used PRISMA criteria.

This review is based only on 9 studies and 49% of reported subjects are from one study. This shows that there is simply too little material to review.

I do not think that using “tinnitus” in the title is justified as there are even less information on connection of CMV with tinnitus than with hearing loss in general. Furthermore the authors say that there was no reported tinnitus in any of the studies that they included.

I am certain that JCM is not the place for such study. It is more suited to more specialized journal related to hearing. However, I do not think that we need this study at all. Nowadays, It is fairly simple to seek databases for some specific information. We do not need a dedicated review paper for each subtopic. The authors certainly not convinced me that we need a review on this topic. If the authors are interested in this topic they should do their own research. They even conclude that: “A large prospective cohort study in adults and children with confirmed CMV infection is needed to demonstrate the potential role of acquired CMV infection in the development of SNHL and tinnitus”. I would like to encourage the authors to perform such study  and to use parts of  this manuscript as an introduction instead of standalone review paper.

Minor remarks:

For unknown reasons there are parts of the text highlighted by yellow color.

CMV in the title – it is better to use full name

Line 152: Fisher´s exact – add test

Line 187: study – studies

Line 195: ELISA – please expand when first introduced

Round 2

Reviewer 1 Report

Previous remarks have been appropriately answered to.

Author Response

Dear Reviewer 1

Thank you for giving me the opportunity to submit a revised draft of my manuscript titled “A systematic review on the association of acquired human cytomegalovirus infection with hearing loss” to Journal of Clinical Medicine. We appreciate the time and effort that you and the others reviewers have dedicated to providing your valuable feedback on my manuscript. We have been able to incorporate changes to reflect most of the suggestions provided by the reviewers. .  All changes in the text have been marked in red

Previous remarks have been appropriately answered to.

Thank you for your useful comments.

Reviewer 2 Report

I think that my advices have been followed, the text is much more fluid and logical. According to me, some minor changes are still needed.

Introduction

In my opinion, only one clarification is missing (a short sentence may be fine), namely that congenital CMV can cause the onset of deafness or progressive deafness in childhood even up to 6-8 years of life

Line 301 : I don’t understand the sentence “It is believed that hearing impairment does not occur in acquired CMV infection [35]” . Maybe is it not the correct context for this statement?.

Line 313: better to say “ although limited”.

Line 341: please substitute “SNHL and tinnitus” with hearing impairment.

Conclusion

Line 338 : This sentence “Our findings suggest that no data supports postnatal CMV-induced SNHL” sounds too peremptory. I think that it should be deleted.

Author Response

Dear Reviewer 2

Thank you for giving me the opportunity to submit a revised draft of my manuscript titled “A systematic review on the association of acquired human cytomegalovirus infection with hearing loss” to Journal of Clinical Medicine. We appreciate the time and effort that you and the others reviewers have dedicated to providing your valuable feedback on my manuscript. We have been able to incorporate changes to reflect most of the suggestions provided by the reviewers. .  All changes in the text have been marked in red

I think that my advices have been followed, the text is much more fluid and logical. According to me, some minor changes are still needed.

Introduction

In my opinion, only one clarification is missing (a short sentence may be fine), namely that congenital CMV can cause the onset of deafness or progressive deafness in childhood even up to 6-8 years of life

We have included the next sentence, according to your suggestion: “Among children with congenital CMV infection, the onset of deafness can occur in childhood even up to 6-8 years of life” (Pag 2, line 65-66).

Line 301 : I don’t understand the sentence “It is believed that hearing impairment does not occur in acquired CMV infection [35]” . Maybe is it not the correct context for this statement?.

We have deleted this sentence.

Line 313: better to say “ although limited”.

Thank you for pointing this out. The revised text reads as follows on: “According to the limited evidence, 60% of individuals without SNHL have antibodies anti-CMV IgG”. (Pag 11, line 327-328).

Line 341: please substitute “SNHL and tinnitus” with hearing impairment.

The substitution has been made on Pag 11, line 358.

Conclusion

Line 338 : This sentence “Our findings suggest that no data supports postnatal CMV-induced SNHL” sounds too peremptory. I think that it should be deleted.

As suggested by the reviewer, we have deleted the sentence.   

Reviewer 3 Report

I believe that use of “tinnitus” in the title is unjustified and unethical. I understand that the authors used this word in the search. But since there are no results there is no justification for use it in the title. The authors may use it throughout manuscript and in the abstract and that should be sufficient.

In my previous review I pointed out that the Experimental section is a little bit chaotic. I do not see any changes in this version. Experimental section should be solely focused on methodology. The place for justification is in the Introduction.

Author Response

Dear Reviewer 3

Thank you for giving me the opportunity to submit a revised draft of my manuscript titled “A systematic review on the association of acquired human cytomegalovirus infection with hearing loss” to Journal of Clinical Medicine. We appreciate the time and effort that you and the others reviewers have dedicated to providing your valuable feedback on my manuscript. We have been able to incorporate changes to reflect most of the suggestions provided by the reviewers. .  All changes in the text have been marked in red

I believe that use of “tinnitus” in the title is unjustified and unethical. I understand that the authors used this word in the search. But since there are no results there is no justification for use it in the title. The authors may use it throughout manuscript and in the abstract and that should be sufficient.

We kindly accept your suggestion. The word “tinnitus” has been removed form the title.

In my previous review I pointed out that the Experimental section is a little bit chaotic. I do not see any changes in this version. Experimental section should be solely focused on methodology. The place for justification is in the Introduction.

We have revised the experimental section to reduce it to the description of the methodology.

This manuscript is a resubmission of an earlier submission. The following is a list of the peer review reports and author responses from that submission.

Round 1

Reviewer 1 Report

Reviewer comments

General Comments

I think this is a worthwhile entity to study however I am concerned that the mechanism by which it has been done is easy to misunderstand.

Would recommend significant rewriting such that the authors clearly distinguish between congenital and acquired impact of CMV.  Throughout the text when CMV infection is mentioned it does require that it be qualified by being either congenital or acquired.  At times the authors use the term non-congenital if this is the term that they chose then it should be used consistently throughout.

The authors should clarify that both the direct and indirect evidence of CMV infection is not time stamped and that at best this is a reflection of prior exposure and thus there are challenges in determining causality with hearing loss.  At best we can expect to examine the prevalence of past CMV exposure in individuals with hearing loss vs. that of the general population.

Specific Comments

Line 170 – The authors state that none of the 77 individuals with CMV infection presented with SNHL however it seems to me that in the methods hearing loss was an exclusion?

3.3. Methods – How was congenital CMV ruled out in this population?  If it wasn’t than that should be stated as a limitation of the study.

Author Response

Dear Reviewer 1

Thank you for reviewing our manuscript entitled "A systematic review on the association of acquired human CMV infection with hearing loss or tinnitus. We have revised it according to the recommendations that you have indicated.  All changes in the text have been marked in red. We sincerely appreciate all the comments and recommendations to improve our article.  We have also answered all the questions point-by-point in order to clarify the contents of the manuscript.

General Comments

I think this is a worthwhile entity to study however I am concerned that the mechanism by which it has been done is easy to misunderstand.

Would recommend significant rewriting such that the authors clearly distinguish between congenital and acquired impact of CMV.  Throughout the text when CMV infection is mentioned it does require that it be qualified by being either congenital or acquired.  At times the authors use the term non-congenital if this is the term that they chose then it should be used consistently throughout.

You are absolutely right.  We have rewritten the text in order to distinguish between congenital and acquired CMV infection. Also, we have change “non-congenital CMV infection” for “acquired CMV infection” throughout the text.

The authors should clarify that both the direct and indirect evidence of CMV infection is not time stamped and that at best this is a reflection of prior exposure and thus there are challenges in determining causality with hearing loss.  At best we can expect to examine the prevalence of past CMV exposure

We agree with your suggestion. We have added this in the text “Evidence of CMV infection (direct and indirect) is not time-stamped, so, to determine causality with SNHL is necessary to examine the prevalence of past exposure in individuals with hearing loss vs. that of the general population”.

 (Page 8. Line 266-268).

Specific Comments

Line 170 – The authors state that none of the 77 individuals with CMV infection presented with SNHL however it seems to me that in the methods hearing loss was an exclusion?

We have modified this in the text: “Individuals were considered as controls if they did not show any evidence of hearing loss, regardless of the past exposure to CMV”.  (Page 3. Line 110-111).

3.3. Methods – How was congenital CMV ruled out in this population?  If it wasn’t than that should be stated as a limitation of the study.

In the three selected studies the keyword “congenital CMV” not appeared in the title, abstract or text Word. All selected cases in these three studies were considered as postnatal acquired CMV infection cases, although congenital CMV infection was not tested during pregnancy in these studies and a latent congenital infection with delayed effects on hearing can not be rule out. We state this as a limitation of the study.  

Reviewer 2 Report

A systematic review on the association of acquired human CMV infection with hearing loss or tinnitus.

Although a systematic review on the topic of acquired CMV and hearing loss would certainly be of interest to many readers of the Journal of Clinical Medicine, I believe that the final selection of articles precludes the authors to reach the level of a meaningful review. I do also see discrepancies between the goal of the review and the 3 articles selected. The authors describe the objective of the study (third sentence of the abstract) as a search of the evidence associating acquired CMV with adult onset of hearing loss and/or tinnitus. Yet, 2 of the 3 articles selected are about CMV infection in preterm infants and young children. Also, the authors decided to exclude studies without a control group. Yet, the study by Paradowska et al (2013) did not include a control group and Wilson study does not accurately define their control group (likely individuals without sudden hearing loss and not specifically without CMV).

I strongly believe that this review could become a noteworthy reference for many clinicians and scientists if a larger number of studies were included to start with by revising the inclusion/exclusion criteria. By excluding several of these studies, the authors are limiting themselves to three very different studies that cannot really help answering their main goal. That goal should also be restated to include infants and children.

Specific comments:

  • The terms “acquired” or “congenital” should always be added to “CMV” throughout the manuscript as some statements made are confusing since we do not know if the authors refer to acquired CMV or congenital CMV (example: line 78-81; 196; 235).
  • Line 51: a reference regarding the seroprevalence of CMV among countries is necessary. Cannon et al., 2010 (Review of cytomegalovirus seroprevalence and demographic characteristics associated with infection) would be one.
  • Line 53: I don’t think the reference provided to describe the effect of aging is appropriate. The study quoted here is about organ donor in Ireland, not patients.
  • Line 158: only 215 subjects have hearing loss, not “all” of them.
  • Line 251-252: if these studies refer to acquired CMV infection and hearing impairment, why are they not part of the review?

Author Response

Dear Reviewer 2

Thank you for reviewing our manuscript entitled "A systematic review on the association of acquired human CMV infection with hearing loss or tinnitus. We have revised it according to the recommendations that you have indicated.  All changes in the text have been marked in red. We sincerely appreciate all the comments and recommendations to improve our article.  We have also answered all the questions point-by-point in order to clarify the contents of the manuscript.

General Comments

A systematic review on the association of acquired human CMV infection with hearing loss or tinnitus.

Although a systematic review on the topic of acquired CMV and hearing loss would certainly be of interest to many readers of the Journal of Clinical Medicine, I believe that the final selection of articles precludes the authors to reach the level of a meaningful review. I do also see discrepancies between the goal of the review and the 3 articles selected. The authors describe the objective of the study (third sentence of the abstract) as a search of the evidence associating acquired CMV with adult onset of hearing loss and/or tinnitus. Yet, 2 of the 3 articles selected are about CMV infection in preterm infants and young children. Also, the authors decided to exclude studies without a control group. Yet, the study by Paradowska et al (2013) did not include a control group and Wilson study does not accurately define their control group (likely individuals without sudden hearing loss and not specifically without CMV).

We agree with your comment. We have not found any study conducted in adults with a proper control group. We have deleted the word “adult” in the abstract and we have modified this in the text: “The objective of this review is to critically appraise scientific evidence regarding the association of acquired CMV infection with postnatal hearing loss and tinnitus”  (Page 1. Line 16-17)

All studies included in this systematic review present a defined control group according to the definition of control group: “Individuals were considered as controls if they did not show any evidence of hearing loss, regardless of the past exposure to CMV”.  (Page 3. Line 110-111)

In Paradowska et al, 2013 the control group consisted of 55 HCMV infected adults, but hearing testing was not reported. Wilson, 1986 described as one hundred twenty-two patients among were 125 seroconversions; (some patients had no seroconversions while others had multiple seroconversions) compared to 95 controls that had 62 seroconversions, but hearing testing was not reported.

I strongly believe that this review could become a noteworthy reference for many clinicians and scientists if a larger number of studies were included to start with by revising the inclusion/exclusion criteria. By excluding several of these studies, the authors are limiting themselves to three very different studies that cannot really help answering their main goal. That goal should also be restated to include infants and children.

We agree with the reviewer and we have modified the goal as stated above to include infants and children. However, the selection of non-control studies will generate a bias and uncertain conclusions. We state the need for new well-designed case-control studies to assess the role of acquired CMV in the postnatal life in the pathogenesis of SNHL.

Specific comments:

  • The terms “acquired” or “congenital” should always be added to “CMV” throughout the manuscript as some statements made are confusing since we do not know if the authors refer to acquired CMV or congenital CMV (example: line 78-81; 196; 235).

Thank you for this suggestion. We have rewritten the text in order to distinguish between congenital and acquired CMV infection. Also, we have change “non-congenital CMV infection” for “acquired CMV infection” throughout the text.

  • Line 51: a reference regarding the seroprevalence of CMV among countries is necessary. Cannon et al., 2010 (Review of cytomegalovirus seroprevalence and demographic characteristics associated with infection) would be one.

We have added the suggested reference to the manuscript on page 2, line 52.

  • Line 53: I don’t think the reference provided to describe the effect of aging is appropriate. The study quoted here is about organ donor in Ireland, not patients.

We have removed the sentence and the associated reference of the manuscript

  • Line 158: only 215 subjects have hearing loss, not “all” of them.

Thanks for this comment and we have it changed in line 158. The new sentence reads as follows: “Among the 409 individuals included, 215 subjects suffered some type of hearing disorder: uni or bilateral hearing loss (mild, moderate, severe or profound) or sudden SNHL.”

(Page 5. Line 157-158)

  • Line 251-252: if these studies refer to acquired CMV infection and hearing impairment, why are they not part of the review?

Thank you for pointing this out. The sentence refers to congenital CVM infection. We have corrected as: “However, some reports have investigated the relationship between congenital CMV infection and hearing impairment [36, 37]”. (Page 8. Line 251-252)
